

# Effectiveness of FIFA injury prevention programs in reducing ankle injuries among football players: a systematic review

Mohammad Alhazmi[1], Emad Alhazmi[2], Wael Abdulghani Alghamdi[3], Mohammed Zalah[1], Shadab Uddin[4], Moattar Raza Rizvi[5] and Fuzail Ahmad[6]

[1] Medical Rehabilitation Center, King Fahad Central Hospital, Jazan, Saudi Arabia
[2] Laboratory and Blood Bank, King Fahd Central Hospital, Jazan, Saudi Arabia
[3] Sports Injuries Rehabilitation Center, Aseer Central Hospital, Abha, Saudi Arabia
[4] Department of Physical Therapy, College of Nursing and Health Sciences, Jazan, Saudi Arabia
[5] College of Healthcare Professions, DIT University, Mussoorie-Diversion Road, Village Makkawala, Dehradoon, Uttrakhand, India
[6] Respiratort Care Department, College of Applied Sciences, Almaarefa University, Diriya, Riyadh, Saudi Arabia

Corresponding author
Fuzail Ahmad, fahmad@um.edu.sa

## ABSTRACT

**Background**. Ankle injuries are some of the most common injuries among footballers and can prevent players from participating in sport. The Fédération Internationale de Football Association (FIFA) developed FIFA injury prevention programs to reduce overall football injuries, but their effectiveness on a particular joint, such as the ankle, was not assessed. Therefore, this project aims to investigate if these FIFA injury prevention programs are particularly effective in reducing ankle injuries.

**Method**. A systematic review was conducted with the following criteria for study selection: Randomized controlled trials (RTCs) comparing the FIFA programs (FIFA 11 the old version and FIFA 11+) with the usual training for the teams. Other studies that included the number or percentage of ankle injuries as an outcome. Sample: Male and female footballers aged between 13 and 40 years old and without any restriction on particular skill levels. The Physiotherapy Evidence Database scale (PEDro) was used to assess the methodological quality of the included studies.

**Results**. The search strategy identified five RCTs that used the FIFA programs. The included studies had a good-to-excellent methodological quality according to the PEDro scale (ranging from 5 to 7 out of 10). The pooled data from all the included studies indicated that the FIFA 11 and FIFA 11+ programs were effective in reducing the ankle injury rate by 14%, while pooled results from the studies that used the FIFA 11+ program significantly reduced ankle injury rate by 32%. However, the FIFA 11 program was not effective in reducing the ankle injury rate.

**Conclusion**. The FIFA programs, and especially the FIFA 11+ program, appear to be more effective than the usual training in preventing ankle injuries among footballers.

## INTRODUCTION

Football is the most common sport worldwide, with nearly 400 million footballers generating roughly $1 trillion annually in more than 200 countries (*Dvorak et al., 2004*). The Fédération Internationale de Football Association (FIFA) has estimated that approximately 270 million international footballers are registered in the world (*Sadigursky et al., 2017*). As in all other sports, football carries an injury risk for all levels of play, professional and amateur, as well as for all age groups. The financial damage incurred by the treatment of football injuries was nearly 118 million EUR in the English professional football leagues over the period from 1999–2000 (*Woods et al., 2002*), while the treatment of football injuries cost Switzerland almost 95 million EUR in 2003 (*Junge et al., 2011*). In the Netherlands, medical costs from football injuries were, amazingly, approximately 1.3 billion EUR in 2008 (*Barengo et al., 2014*).

The ankle joint is particularly susceptible to injuries in football, with ankle sprains being the most common injury across all levels of play, from amateur to professional athletes (*Agel et al., 2007*; *Van Rijn et al., 2008*). Activities such as hopping, landing on one foot, and sudden directional changes frequently lead to sprains, resulting in the highest percentage of ankle injuries among sports (*Chomiak et al., 2000*; *Fong et al., 2007*). Notably, around 80% of athletes experience recurring sprains after an initial injury (*Hale & Hertel, 2005*). During the 2004 Olympics, ankle injuries were more prevalent than in any other sports event (*Walls et al., 2016*), and over two million ankle sprains are treated annually in emergency departments in the United States and the United Kingdom, often linked to football (*Doherty et al., 2017*). Consequently, preventing ankle injuries has become a priority in sports and healthcare (*Grimm et al., 2016*).

Ankle stability largely depends on the integrity of ligaments, particularly the anterior talofibular and calcaneofibular ligaments (*Panagiotakis et al., 2017*). Common injury mechanisms include sudden inversion, internal rotation, and plantarflexion, which compromise joint stability (*Panagiotakis et al., 2017*; *Fong et al., 2009*). Cadaveric studies have further elucidated these injury patterns (*Funk, 2011*). Biomechanical factors, such as foot type and muscle activation, play a role in injury risk, and chronic instability may arise from incomplete ligament healing or impaired proprioception. A thorough understanding of ankle biomechanics is essential for developing effective injury prevention programs, guiding clinical treatment, and enhancing rehabilitation outcomes (*Alhammad et al., 2024*).

Numerous scientific studies have been published on injury prevention programs for footballers (*Bizzini & Dvorak, 2015*). The sports community still faces significant challenges in implementing an effective injury prevention program (*Bizzini & Dvorak, 2015*). Effective prevention strategies include structured warm-ups, strength training, proprioceptive exercises, and balance training (*Pérez-Gómez et al., 2022*). Preseason conditioning, functional training, education, balance exercises, and sport-specific skills training have shown effectiveness in reducing injuries (*Abernethy & Bleakley, 2007*). While stretching is commonly recommended, its role in injury prevention remains unclear (*Stojanovic & Ostojic, 2011*).

However, research has shown that nationwide strategies can reduce football injuries (*Gatterer et al., 2012*). As an example, the FIFA 11 injury prevention project was launched in 2003 with the goal of reducing overall football injuries, with the supervision of the FIFA Medical Assessment and Research Centre (F-MARC) (*Al Attar et al., 2016*). After a workout, one should use FIFA 11's warm-up routine, which includes ten exercises and takes about fifteen minutes to finish (*Al Attar et al., 2016*). In a Norwegian study (*Steffen et al., 2008*), researchers looked at how well the FIFA 11 program reduced football injuries generally. The results showed that the curriculum had no effect on female football players. A similar study conducted in the Netherlands by *Van Beijsterveldt et al. (2012)* found no significant impact of the FIFA 11 program on lowering the total occurrence of football injuries among male amateurs. However, thus far no research has looked at how it affects ankle joints in particular (*Van Beijsterveldt et al., 2012*).

The FIFA 11+ project was introduced in 2006 as a replacement for FIFA 11, which had undergone upgrades in the previous year. A group of sports scientists and experts worked on FIFA 11+ to make it more effective in reducing football injuries, especially those that harm the lower limbs (*Bizzini & Dvorak, 2015*). This course offers a total of 15 structured exercises, may be accessed online, and is straightforward to perform (*Bizzini & Dvorak, 2015*). The exercises consist of eccentric training of the thigh muscle, core stabilization, proprioceptive training, dynamic stabilization, and plyometric exercises (*Sadigursky et al., 2017*). The FIFA11+ program has been proven to significantly decrease the total risk of football injuries by as much as 50% (*Al Attar et al., 2016*). Nevertheless, there has not been a comprehensive analysis that specifically examines a specific region of injury, such as the ankle. This systematic review aims to examine the impact of the FIFA injury prevention programs (namely FIFA 11, the older version, and FIFA 11+) on ankle injuries in football players aged 13–40 years, regardless of gender.

The FIFA 11 and FIFA 11+ programs are multi-component, evidence-based injury prevention routines designed specifically for football players (*Al Attar et al., 2016*). Unlike general injury prevention strategies, these programs replace typical warm-ups and target common football injuries, including those to the ankle. The FIFA 11 program includes 10 exercises focused on core stability, muscle strength, and dynamic balance, completed in about 10–15 min (*Van Beijsterveldt et al., 2012*). The updated FIFA 11+ program adds more exercises and progression for increased effectiveness, with 15 exercises emphasizing proprioception, stability, and running mechanics, completed in about 20 min (*Thorborg et al., 2017*). The FIFA 11+ program's structured progression and added exercises have demonstrated increased effectiveness in reducing football injuries, making it particularly valuable for preventing injuries at the ankle—a common site of injury in the sport. By focusing on FIFA 11 and FIFA 11+, our study highlights the potential of these evidence-based, sport-specific programs to improve player safety and reduce injury-related healthcare costs globally. We believe this targeted focus provides actionable insights for football practitioners, coaches, and policymakers aiming to implement effective, scalable injury prevention protocols tailored to football.

*Barengo et al. (2014)* reported that with the FIFA 11+ program, injuries to footballers decreased by 30–70%. However, this study was not published in a journal with high impact

and the review had some methodological limitations. Some of the included studies have demonstrated low compliance with the intervention which is a matter of concern as it directly affects the outcomes (*Van Reijen et al., 2016*). Moreover, it included explorative and observational cohort studies that are considered to provide a low level of evidence, such as *Evans (2003)* and the studies included multiple outcome measure making it prone to bias.

In contrast the systematic review by *Al Attar et al. (2016)* including nine studies with three different FIFA prevention programs compared the effects of the FIFA injury prevention programs with the usual training of teams. The findings of this systematic review demonstrated that the FIFA injury prevention programs reduced overall football injuries and lower extremity injuries by 23% and 24% respectively, compared to the control group. The limitations of this review were inclusion of cohort studies which provided low level of evidence (*Evans, 2003*) with heterogeneous sample sizes (71–2,020 participants), and different interventions with varied duration (12–40 weeks), which could have negatively affected the review's results. The included studies failed to mention the components of the teams' usual training exercises, thereby making it impossible to compare the effects of these standard exercises with those of the FIFA injury prevention programs.

The recent systematic review conducted by *Sadigursky et al. (2017)* examined the effect of the FIFA 11+ injury prevention program and reported that in football players of both genders, injuries were reduced by 30%. However, one of the limitations of this review was the age of the participants, which ranged from 13 to 25 years. However, a study (*Hammes et al., 2015*) in which the participants were overweight and aged more than 40 years and another study (*Hulin et al., 2014*) that applied the intervention for different lengths of time were included. Another limitation was that the systematic review reported that males experienced a greater reduction in the rate of injury than females, but there was no clear explanation for why this occurred. These differences meant that the participants had differing exposures to the intervention and can increase the likelihood of injury when playing sport and may have distorted the review's results and made it difficult to compare them.

The meta-analysis by *Thorborg et al. (2017)* reported a reduction of 39% in overall football injuries with the use of the FIFA 11+ injury prevention program. Looking at particular injuries, this program reduced the three most common football injuries (knee, ankle, and hip injuries) by 48%, 32%, and 41% respectively. This is extremely useful as sports practitioners can utilize this information to target specific injuries. However, as with *Sadigursky et al. (2017)* and *Hammes et al. (2015)*, this could distort the results of the review because it incorporated older and overweight participants. Two further limitations of this review were that it did not determine which exercises were the most effective in preventing injuries to the lower limbs, and it failed to make recommendations regarding how often or how intensively the exercises should be implemented.

The systematic review conducted by *Gomes Neto et al. (2017)* included eleven RCTs and showed a significant reduction in the risk of injury among football players, as well as an improvement in their agility and dynamic balance, with the FIFA 11 program. However, the review limited itself to general injuries and did not examine the effectiveness of the FIFA

11 program for injuries in particular parts of the body. One of the benefits of this review, however, was that it examined how the FIFA 11 program affected agility and dynamic balance. This is important because if balance is impaired, this can increase the risk of knee and ankle injuries (*Emery et al., 2005*).

It is widely recognized that the FIFA injury prevention program are comprehensive programs that help to avoid overall football injuries among footballers (*Al Attar et al., 2016*). The majority of previous studies have highlighted how the FIFA 11+ program can reduce football injuries to the lower and upper limbs as well as overall injury incidence rates (*Mayo, Seijas & Alvarez, 2014*). While research has shown the program to be effective in reducing general lower limb injuries, researchers have not provided enough evidence to help coaches decide which specific injuries (*e.g.*, ankle or knee injuries) are best targeted by the FIFA 11+ program (*Mayo, Seijas & Alvarez, 2014*).

The literature search has revealed a significant gap as no previous systematic reviews have focused on how the FIFA injury prevention programs can help to prevent ankle injuries. Furthermore, in terms of injuries to the ankle, ankle sprains are the most common type (*Fong et al., 2007*); therefore, investigating a program that may help to prevent them has great value in this field. It is clear that focusing on this type of injury will help to reduce the incidence of ankle injuries among football players, particularly those who have experienced these injuries in the past. The severity of these injuries for football players can take a long time to recover and be able to rejoin the team (*Walls et al., 2016*). As the FIFA prevention programs includes proprioception (balance) exercises (*Al Attar et al., 2016*) which can help to prevent ankle injuries, and they therefore make the FIFA programs interesting to study in terms of reducing the incidence of ankle injuries among sportsmen.

The aim of this study is to investigate if the FIFA injury prevention programs (FIFA 11 and FIFA 11+) are effective in reducing ankle injuries in football players between 13 and 40 years of age, through a systematic review of contemporary literature. Specific objectives for our study include, finding the effectiveness of the FIFA injury prevention programs by comparing the incidence of ankle injuries between intervention and control groups and to investigate if the FIFA injury prevention programs are specifically effective in reducing ankle sprains rather than other ankle injuries.

## METHODOLOGY

The systematic review was registered in Prospero on March 4th, 2024, with the registration number CRD42024503486.

### Search strategy

Five databases were used to ensure all the related articles were included in this systematic review, as well as to ensure and test the reliability and quality of the included studies. We limited our search to studies published from 2008 to 2024 to capture research conducted after the introduction of the FIFA 11+ program, ensuring relevance to current injury prevention strategies. Although only randomized controlled trials (RCTs) were included to maintain methodological rigor, we acknowledge that excluding cohort studies-commonly used in sports injury research-may limit the scope of this review. Future reviews may

benefit from incorporating high-quality cohort studies to broaden the evidence base for injury prevention (*Soligard et al., 2008*). However, the traditional way of searching, namely, a manual search, was also used. The databases searched were MEDLINE (Ovid), PubMed, CINAHL, Scopus and Cochrane Central Register of Controlled Trials (Cochrane library).

The following keywords were used to carry out the electronic searches: ''injury prevention'' or ''FIFA 11+'' or ''FIFA 11'' or ''The 11+'' or ''FIFA 11+ warm-up program'' or ''neuromuscular training'' and ''football'' or ''soccer'' and ''sport injuries''. The search was conducted by the author from February 2024 to July 2024. Initially, identified articles from the search strategy were uploaded and duplicates were removed, using EndNote. The injury incidence rate or number of injuries to the ankle joint per 1,000 h of football exposure were taken as the outcome measures for this study.

Criteria for selecting studies included only RCTs (level I evidence) to avoid bias and to produce a high quality, systematic review (*Evans, 2003*). Furthermore, studies that included ankle injury incidence rates or numbers in their outcome measures were eligible for inclusion. Studies that used either the FIFA 11 or the FIFA 11+ injury prevention programs as interventions (see Table 1) were also included in the study.

## Study selection

The study selection process aimed to include all the relevant trials in the systematic review (*Tacconelli, 2010*). The study selection was completed two researchers independently reviewing the eligibility of the trials and to ensuring their reliability, thereby reducing bias (*Tacconelli, 2010*). The returned studies were screened using the inclusion criteria from the database searches based on the title and the abstract. However, in the case of studies where a decision could not be taken, a full text was obtained to facilitate the decision for their inclusion. After completing the initial screening of the returned articles, complete articles were collected from those that met the inclusion criteria that had been determined by a comprehensive assessment of such criteria. Endnote was utilized for handling references and making decisions on exclusion factors.

## Data extraction

The data extracted from that was used in this systematic review; it was adapted from the study by *Thorborg et al. (2017)*. To prevent bias, Mohammad Alhazmi collected the data while Mohammed Zalah independently tested the consistency of, and the information provided by the data extraction process (*Tacconelli, 2010*). If any discrepancies arose between the evaluators, they were resolved through consensus and arbitration according to a predefined strategy, with Fuzail Ahmad acting as the referee to resolve any disagreements (*Tacconelli, 2010*). Data were extracted from the five eligible studies that met the inclusion criteria which encompassed study design, participant characteristics, intervention compliance, intervention types, and outcome measures, specifically ankle joint injury incidence. Missing data were sought by contacting study authors *via* email.

## Quality assessment

As this systematic review includes only randomized controlled trials, the PEDro scale was used to assess the quality of the studies involved in this systematic review. The PEDro

**Table 1** A summary of the characteristics of the studies included in this systematic review.

| Study | Number of participants | Age (years) | Study design | Duration of programme (IG) | Duration of programme (CG) | Frequency of programmes both IG & CG |
|---|---|---|---|---|---|---|
| *Silvers-Granelli et al. (2015)* (FIFA 11+) | 1,525 male football players (IG 675) and (CG 850). | 18–25 | RCT | 6 months of the FIFA 11+ programme. | 6 months of the usual team | 20 min of exercise, once/week |
| *Soligard et al. (2008)* (FIFA 11+) | 1,892 female young footballers (IG 1055) and (CG 837). | 13–17 | RCT | 8 months of the FIFA 11+ programme. | 8 months of the usual team training | 20 min of exercise, 3 times/week |
| *Owoeye et al. (2014)* (FIFA 11+) | 416 male footballers (IG 212) and (CG 204) | 14–19 | RCT | 6 months of the FIFA 11+ programme. | 6 months of the usual team training | 20 min of exercise, once/week |
| *Van Beijsteveldt et al. (2012)* (FIFA 11) | 456 male footballers (IG 223) and (CG 233) | 18–40 | RCT | 9 months of FIFA 11 programme. | 9 months of the usual team training | 10–15 min of exercise, twice/week |
| *Steffen et al. (2008)* (FIFA 11) | 2,020 female young footballers (IG 1073) and CG 947). | 13–17 | RCT | 8 months of FIFA 11 programme. | 8 months of the usual team training | 10–15 min of exercise, once/week |

**Notes.**

Key: G, Group; I, Intervention; C, Control; Ex, Exercises and RCT, Randomised Control Trial (Used as abbreviations).

scale, based on the Delphi list and renewed by *Verhagen et al. (1998)*, is a quality evaluation framework for randomized clinical trials to report systematic reviews (*Mokkink et al., 2010*). It focuses on covert allocation, intention-to-treat analysis, and follow-up adequacy, making it an effective tool for assessing the quality of physiotherapy and rehabilitation studies (*Olivo et al., 2008*). The PEDro scale consists of 11 items, but one component (eligibility criteria) pertains to external validity and is not commonly used to measure the method score, resulting in a score range of 0 to 1 (*Maher et al., 2003*). The PEDro scale was used to assess study quality, with scores ranging from 5 to 7 among the included studies. Lower scores were primarily due to methodological limitations such as lack of blinding (both participants and trainers) and inadequate follow-up periods. Given the nature of exercise intervention studies, blinding was challenging, as trainers were aware of the intervention being implemented, which may introduce performance bias. Inadequate follow-up periods in some studies could also limit the robustness of long-term outcomes. The assessment of the quality of the included studies should is performed by two independent reviewers to avoid bias (*Tacconelli, 2010*).

## Data synthesis

For this project, a systematic review with a narrative synthesis was chosen instead of a meta-analysis due to the high heterogeneity among the included studies. To assess the potential for meta-analysis, we calculated the $I^2$ statistic, which indicated substantial variability (>75%) among studies. This variability arose from significant differences in participant demographics and intervention protocols. Participants varied across studies, including adolescent females, adolescent males, and adult males, reflecting differences in biological maturity that could impact ankle injury incidence. Additionally, heterogeneity was evident in the types of FIFA injury prevention programs used, with two studies

employing the original FIFA 11 program and three utilizing the modified FIFA 11+ program. These versions differ in warm-up duration, exercises included, and progression, further contributing to study variability. Given these factors, pooling effect sizes in a meta-analysis could produce biased estimates and limit result interpretability. Consequently, a narrative synthesis was selected to provide a more nuanced interpretation of findings across these diverse study contexts.

### Injury definitions, injury rates, and the injury rate ratio

In football, an injury is defined as one that prevents players from participating in the next game and training sessions (*Al Attar et al., 2016*). The injury rate (IR), a key measure in sports research, is calculated by dividing the number of injuries by the total exposure hours (training and games) and multiplying by 1,000 (*Caine, Maffulli & Caine, 2008*) was used to compare ankle injury incidence between intervention and control groups. The injury rate ratio (IRR), calculated by dividing the IR of the intervention group by the IR of the control group, indicates intervention effectiveness; an IRR less than 1 shows a positive effect (*Al Attar et al., 2016*). This review used the IRR to demonstrate the effectiveness of the FIFA 11 and FIFA 11+ injury prevention programs in reducing ankle injuries. The total hours of football exposure were also used in this systematic review to compare the findings from the intervention group *versus* those from the control group. The difference in compliance with the FIFA prevention programs (FIFA 11 and FIFA 11+), which was measured by calculating the number of weekly training sessions for each team and player (*Thorborg et al., 2017*) between the various studies in this systematic review was used to document the relationship between compliance to the prevention programs and the actual ankle injury reduction rate.

### Identified studies

The five electronic databases (MEDLINE (Ovid), PubMed, CINAHL, Scopus, and Cochrane Central Register of Controlled Trials) yielded 640 articles. After the removal of duplicates (150 articles), the remaining 490 articles were excluded after screening titles (420) and abstracts (50). As a result of this culling, 20 articles were obtained for full-text review. Of those, 15 out of the 20 were excluded because they failed to meet the inclusion criteria; for example, they were studies that were conducted on other sports (not football), studies that did not include the FIFA 11 or FIFA 11+ prevention programs in the intervention group, and studies that were not RCTs. A full description of the whole studies identification process is provided in Fig. 1.

## RESULTS

### Descriptive analysis of the identified studies

A summary of all the studies that were included in this systematic review is available in Table 1. There was a total of five studies, all of which were RCTs. These studies were conducted in four different countries. Two of them were in Norway (*Funk, 2011*; *Evans, 2003*), one in the Netherlands (*Van Beijsterveldt et al., 2012*), one in the USA (*Silvers-Granelli et al., 2015*) and one in Nigeria (*Owoeye et al., 2014*).
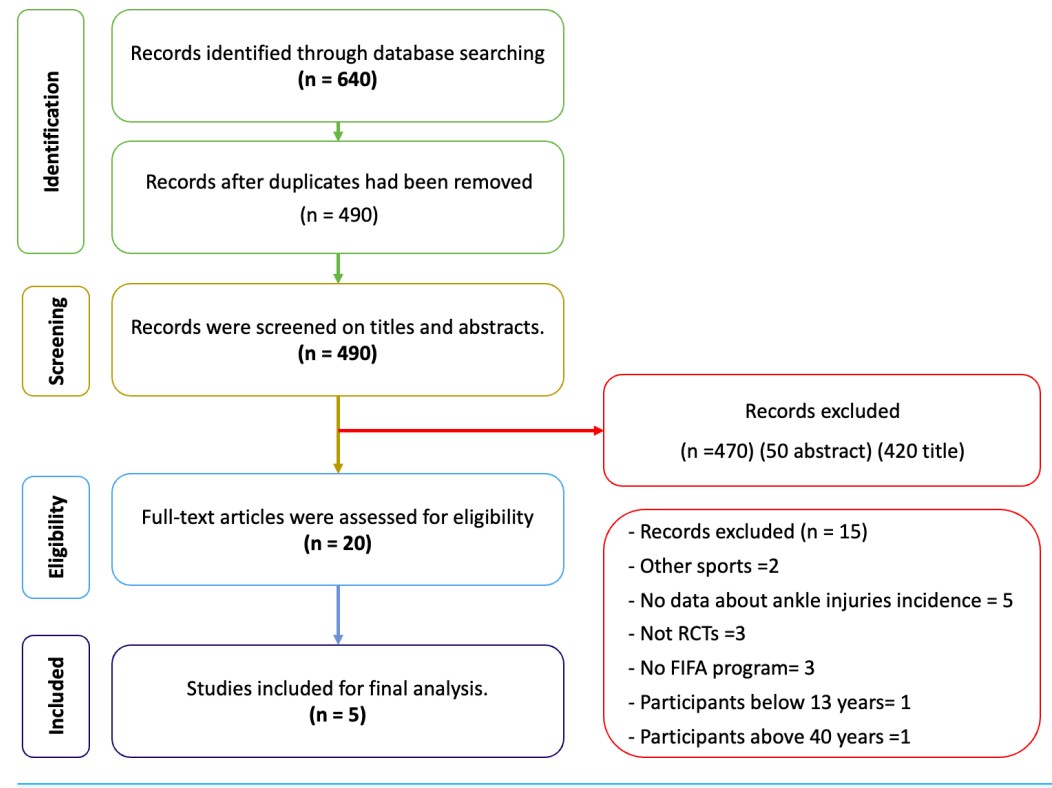

**Figure 1** A PRISMA flow diagram of the article selection process of the systematic review.

## Sample characteristics

The five included RCTs randomly allocated 158 teams (3,238 players) to the intervention group (the FIF 11 and FIFA 11+ injury prevention programs) and 148 teams (3,071 players) to the control group (the usual training for the team). The mean age of the players in the included studies ranged from 13 to 40 years old. The five included RCTs in three studies which had male footballers (*Alhammad et al., 2024*; *Soligard et al., 2008*; *Tacconelli, 2010*) and in two studies that had female footballers (*Funk, 2011*; *Evans, 2003*). Furthermore, none of the included studies had professional football players as a sample; they were either sub-elite or recreational football players (see Table 1).

## Types of intervention

Three of the included studies used the FIFA 11+ injury prevention program in the intervention group and two of them used the FIFA 11 prevention program in the intervention group (FIFA 11 is the old version of the FIFA 11+ program; they have minor differences in the components of the exercises, such as the progression levels of the exercises, which can be found only in the FIFA 11+ program) (*Thorborg et al., 2017*). While all the five studies used the normal or usual training for the teams in the control group to compare it with FIFA 11 and FIFA 11+ programs. none of the included studies exactly explained the components of the usual training for their teams. The intervention duration of the included studies ranged from 5 to 9 months. The frequency of the warm-up

or training programs in both the intervention and control groups ranged from 1 to 3 times a week (see Table 1).

All the included studies in this systematic review had a good-to-excellent methodological quality, according to the PEDdro scale, ranging from 5 to 7. The scores for each study depended on the study's methodology (see Table 2). The PEDro scale measures the study by giving one point (Yes) for each element that has been covered appropriately by the study. Then the total number of points out of 10 is calculated.

There was a big difference in the ankle injury rates per 1,000 h of exposure and in the total ankle injuries number in the five studies included in this systematic review. Table 3 shows the number of ankle injuries, the ankle injury rates per 1,000 of exposure and the total exposure hours for each study.

## Exposure hours in the intervention groups and in the control groups

Of the football players in the intervention teams, 158 had exposure to a total of 241,755 h (the FIFA 11+ 153,755 h and FIFA11 87,985 h), while those in the control teams had exposure to a total of 261,016 h. The difference between the intervention group and the control group in terms of the total hours of football exposure (trainings and matches) could have affected the overall results of this systematic review. This possibility was discussed in the discussion chapter. Table 3 provides a summary of the total exposure hours of each included study in this systematic review.

## The effect of the FIFA injury prevention programs (FIFA 11 and FIFA 11+) on ankle injury reduction

The five included studies that used the FIFA 11+ and FIFA 11 injury prevention programs in the intervention group included (Funk, 2011; Alhammad et al., 2024; Evans, 2003; Soligard et al., 2008; Tacconelli, 2010). The results demonstrated an 14% ankle injury reduction per 1,000 h of exposure in the intervention group, compared to the control group. The ankle injury rate in the intervention groups was 1.0 per 1,000 h of exposure and 1.16 per 1,000 h of exposure for the control groups. Players in the FIFA 11 and FIFA 11+ groups had 244 ankle injuries in 241,740 h, compared to those in the control group who had 304 ankle injuries in 261,016 h (see Table 4).

## The effect of the FIFA 11+ program on ankle injury reduction

Interestingly, the studies that used the FIFA 11+ program in the intervention group (Evans, 2003; Soligard et al., 2008; Tacconelli, 2010) showed a significant reduction in ankle injures incidence of 32% compared to the control group. The ankle injury rates in the FIFA 11+ groups were 0.78 per 1,000 h of exposure compared to the control groups which were 1.14. Football players in the FIFA 11+ group had 120 ankle injuries in 153,755 h compared to those who were playing in the control groups who had 197 ankle injuries in 172,611 h (see Table 4).

## The effect of the FIFA 11 program on ankle injury reduction

In contrast, the results pooled from the studies that used the FIFA 11 program in the intervention group (Funk, 2011; Alhammad et al., 2024) indicated that FIFA 11 (old

**Table 2  Methodological quality scores of studies according to the PEDro scale.**

| Study | Silvers-Granelli et al. (2015) | Soligard et al. (2008) | Steffen et al. (2008) | Owoeye et al. (2014) | Van Beijsteveldt et al. (2012) |
|---|---|---|---|---|---|
| Eligibility criteria | 1 | 1 | 1 | 1 | 1 |
| Random allocation | 1 | 1 | 1 | 1 | 1 |
| Concealed allocation | 0 | 0 | 1 | 0 | 0 |
| Baseline comparability | 1 | 0 | 0 | 1 | 1 |
| Blind subjects | 0 | 0 | 0 | 0 | 0 |
| Blind therapists | 0 | 0 | 0 | 0 | 0 |
| Blind assessors | 1 | 1 | 1 | 0 | 0 |
| Adequate follow-up | 1 | 0 | 1 | 1 | 1 |
| Intention-to-treat analysis | 1 | 1 | 1 | 1 | 0 |
| Between- group comparisons | 1 | 1 | 1 | 1 | 1 |
| Point estimates and variability | 1 | 1 | 1 | 1 | 1 |
| Maximum score | 10 | 10 | 10 | 10 | 10 |
| Study score | 7 | 5 | 7 | 7 | 5 |

Notes.
Key: 1, Yes; 0, No.

**Table 3  Injury rates in both control and intervention groups of the included studies per 1,000 h.**

| Included studies | Ankle injury (IG) | Exposure hours | Ankle injuries per 1,000 h | Ankle injuries (CG) | Exposure hours | Ankle injuries per 1,000 h |
|---|---|---|---|---|---|---|
| Silvers-Granelli et al. (2015) (FIFA 11+) | 59 | 52,839 | 1.1 | 115 | 66,318 | 1.7 |
| Soligard et al. (2008) (FIFA 11+) | 51 | 49,899 | 1.0 | 52 | 45,428 | 1.14 |
| Owoeye et al. (2014) (FIFA 11+) | 10 | 51,017 | 0.19 | 30 | 61,045 | 0.49 |
| Van Beijsteveldt et al. (2012) (FIFA 11) | 45 | 21,562 | 2.0 | 33 | 22,680 | 1.4 |
| Steffen et al. (2008) (FIFA 11) | 79 | 66,423 | 1.18 | 74 | 65,725 | 1.12 |

Notes.
Key: IG, intervention group, and CG, control group.

**Table 4  Injury rates in the studies that used FIFA 11+ and FIFA 11 in the intervention groups to compare with the control groups.**

| Studies | Total number of ankle injury (IG) | Total exposure hours | Ankle injuries per 1,000 h | Total number of ankle injury (CG) | Total exposure hours | Ankle injuries per 1,000 h |
|---|---|---|---|---|---|---|
| FIFA 11+ | 120 | 153,755 | 0.78 | 197 | 172,611 | 1.14 |
| FIFA 11 | 124 | 87,985 | 1.4 | 107 | 88,405 | 1.2 |
| Both FIFA 11+ & FIFA 11 | 244 | 241,740 | 1.0 | 304 | 261,016 | 1.16 |

version) was not effective in reducing ankle injuries in football players. The ankle injury rate in the FIFA 11 group was a higher 1.4 per 1,000 h of exposure compared to the control groups which were only 1.12. Furthermore, footballers in the FIFA 11 group had more ankle injuries (124) in a total of 87,985 h compared to the control groups which were 107 in 88,405 h (see Table 4).

### The effect of the FIFA 11 and FIFA 11+ programs on ankle sprain injuries

This systematic review also aimed to scope types of ankle injuries, such as ankle sprain, that have been associated with having a preventive effect on the FIFA 11 and FIFA 11+ programs. However, the four included studies did not report the types of ankle injuries, and they reported only the total of ankle injuries which could have been fractures or ankle sprains *etc*. An exception was the study by *Steffen et al. (2008)* (FIFA 11), which reported that ankle sprain was the most prevalent (28%) acute injury in both groups (intervention and control group) of the overall acute injuries (421 in total). This study also found that ankle sprain was the most frequent re-injury, with no major difference between both groups (the intervention group had 23 re-injuries and the control group 22 re-injuries). Therefore, these findings indicate that FIFA 11 was not effective in reducing ankle sprain. However, according to the available data in the included studies that used the FIFA 11+ program as an intervention, it is still not clear if FIFA 11+ was effective in reducing ankle sprain.

### Compliance with injury prevention programs (FIFA 11+ and FIFA 11)

The number of training sessions that were performed by teams in both the intervention and the control groups ranged from one to three times a week (see Table 1 above). Compliance with the FIFA prevention programs in the included studies ranged from 46% to 77%. For the control group, however, no data, such as the number of weekly training sessions, was provided about their compliance with the usual training.

The findings from the present systematic review indicate that the FIFA 11+ is indeed an effective injury prevention program for ankle injuries in football players. However, the FIFA 11 program (the old version) was not as effective in reducing ankle injuries in football players as the new 11+ program.

## DISCUSSION

The main finding from this systematic review was that the five included studies that used the FIFA 11 and FIFA 11+ injury prevention programs (*Funk, 2011*; *Alhammad et al., 2024*; *Evans, 2003*; *Soligard et al., 2008*; *Tacconelli, 2010*) as interventions were indeed effective in reducing ankle injury incidence rates. The pooled data from the included studies in fact indicated an 14% reduction in ankle injury rates in the intervention group compared to the control group. The ankle injury rate per 1,000 h of exposure in the intervention group was 1.0 compared to that of the control group that was 1.16 per 1,000 h of exposure.

The results of this systematic review, however, indicated that the FIFA 11 program was not effective in reducing ankle injury incidence when the data were pooled separately compared to the FIFA 11+ program (new version) which indicated a significant 32% reduction in the ankle injury incidence rate. The results of this systematic review concur with previous meta-analyses which concluded that the FIFA 11 program was in fact not effective in reducing lower limb injuries: this included ankle injuries (*Thorborg et al., 2017*).

Based on their findings, *Steffen et al. (2008)* concluded that limited compliance with the FIFA 11 program (53% of participants) was the major reason the program failed

to minimize ankle injuries. On the other hand, *Van Beijsterveldt et al. (2012)* observed no significant reduction in ankle injuries even with a higher compliance rate of 73%, suggesting that compliance alone may not fully account for the program's limitations.

Studies reviewed highlight the FIFA 11+ program's effectiveness in reducing injuries across various anatomical regions, particularly the knee, ankle, and hamstring. *Silvers-Granelli et al. (2015)* found significant reductions in anterior cruciate ligament (ACL) injuries in the knee joint, with the intervention group reporting fewer ACL injuries (3 *vs.* 16) compared to the control group. *Silvers-Granelli et al. (2015)* also observed fewer knee and hamstring injuries in the FIFA 11+ group, with ankle injuries being most prevalent overall. In another study by *Soligard et al. (2008)*, focused on female youth players, the FIFA 11+ program similarly reduced total injuries, with the majority affecting the ankle and knee joints. These results underscore the FIFA 11+ program's targeted benefits for high-risk anatomical regions, specifically supporting knee, ankle, and hamstring injury prevention.

This study fills an important gap in sports injury prevention literature by focusing specifically on ankle injuries in football, a common yet under-researched area in targeted injury prevention. Previous systematic reviews have primarily examined general injury prevention outcomes, while this research isolates ankle injuries, offering unique insights into the efficacy of the FIFA 11+ program for this particular injury type. This specificity provides practical value for coaches and sports practitioners seeking tailored interventions for ankle stability and injury prevention in football. Future research could benefit from replicating these findings in different populations, age groups, and playing levels to evaluate the broader applicability of the FIFA 11+ program. Such research would advance injury prevention protocols and promote safer sports participation across various contexts.

A systematic review on hamstring strain injury prevention in professional and semi-professional football players by *Biz et al. (2021)* finds that various preventive interventions, including the FIFA 11+ protocol, eccentric exercises like Nordic Hamstring Exercises, balance training, and core stability exercises, are effective in reducing injury incidence. The study emphasizes that modifications to the FIFA 11+ program, such as performing specific components at the end of training, could enhance its effectiveness. While the review highlights the potential of these interventions, it acknowledges challenges in assessing the absolute superiority of one protocol over another due to variability in sample sizes and study designs. This aligns with our study, which demonstrates the significant effectiveness of the FIFA 11+ program in reducing ankle injuries, reinforcing its role as a critical preventive tool in football. Despite limitations in pooling data from older versions of the program, our findings and those of *Biz et al. (2021)* advocate for the implementation of tailored FIFA 11+ protocols as a safe and effective strategy to prevent injuries in football players.

The success of the FIFA 11+ program has been linked to its emphasis on correct posture and neuromuscular control during exercises, as well as its inclusion of more intense, longer, and progressive workouts (*Fong et al., 2009*; *Al Attar et al., 2016*). Ankle sprains are more common in football players with poor static and dynamic balance, and according to *Soligard et al. (2008)*, the FIFA 11+ program is better at reducing the likelihood of these injuries than FIFA 11 by improving players' control and coordination of their movements.

On the other hand, etiological aspects and injury processes were not addressed in FIFA 11, and the program had little to no physiological impacts on running speed, agility, or balance (*Funk, 2011*; *Alhammad et al., 2024*).

Furthermore, inconsistencies in the results might be explained by variances in football player attributes between the FIFA 11 and FIFA 11+ studies. Injury occurrence patterns and injury risk factors are affected by sex, age, and levels of play as well as different weather conditions (*Orchard et al., 2013*). The preventive effect may vary in various populations since the FIFA 11 and the FIFA 1+ programs have a group of exercises that make it difficult to decide which particular parts of these programs are effective in preventing lower limb injuries and in reducing ankle injuries. In other words, it is not clear exactly which exercise had reduced the ankle injury rate. Therefore, a comparison of the effectiveness of the FIFA 11 and FIFA 11+ programs through high-quality studies such as RCTs are still needed.

### Exposure hours and ankle injury incidence rate

The literature indicates a significant relationship between exposure hours and athletes' injuries, with increased training and match time leading to greater vulnerability to injury risk factors like tiredness and overload (*Fong et al., 2009*; *Stojanovic & Ostojic, 2011*). Due to the 19,276-h difference, the risk of injury may have been reduced by teams utilizing FIFA 11 and FIFA 11+, since they had fewer exposure hours (241,740) compared to the control group (261,016). In order to reduce the increased risk of injury, elite players, who often play more matches, must carefully control their exposure hours (*Soligard et al., 2008*).

### Delivery of the FIFA injury prevention programs

The ways in which football players are taught workouts are crucial to the success of the FIFA injury prevention programs. According to research (*Sadigursky et al., 2017*; *Tacconelli, 2010*), individuals benefit more from receiving direct supervision from trained professionals like physiotherapists or strength coaches rather than from relying on instructional websites or videos. In comparison to self-teaching techniques, educational workshops with practical demonstrations and constant supervision lead to greater compliance and considerably lower injury rates (*Steffen et al., 2008*). *Steffen et al. (2008)*, for instance, discovered that injury rates were lower, and compliance was higher in comprehensive teams that had both educational workshops and regular physiotherapist supervision (85.6% *vs.* 81.3% in the control group and 73.5% in the workshop group). A standardized protocol with comprehensive training and frequent follow-up would improve the program's preventative benefits, as the program's efficacy can be affected by a lack of standardization and monitoring in delivery techniques (*Evans, 2003*; *Soligard et al., 2008*).

### Compliance with the FIFA injury prevention programs

According to *Silvers-Granelli et al. (2015)* and *Steffen et al. (2008)*, football injury rates are inversely related to compliance with the FIFA 11+ program. Teams with high compliance rates had the lowest incidence of lower limb injuries. The prevalence of ankle injuries was not, however, associated with high compliance with the FIFA 11+ or FIFA 11 programs, according to pooled data. *Owoeye et al. (2014)* observed a lower ankle injury rate (0.19 per 1,000 h) and a greater compliance rate (77%) with one training session per week, in

contrast to *Soligard et al. (2008)*, who reported a higher ankle injury rate (1.0 per 1,000 h) with three sessions per week. Also, despite varying degrees of compliance (53 and 73 percent, respectively), neither *Steffen et al. (2008)* nor *Van Beijsterveldt et al. (2012)* found a statistically significant drop in the incidence of ankle injuries. According to *Thorborg et al. (2017)*, further high-quality research is required to determine the exact nature of the connection between the frequency of training sessions, adherence to FIFA injury prevention programs, and the incidence of football injuries, specifically ankle injuries.

## Intervention durations of the FIFA injury prevention programs

The length of the interventions in the studies that were included ranged from six to nine months. In order to protect, strengthen, and stabilize skeletal joints, injury prevention exercises rely on long-term factors such as muscle strength (*Sadigursky et al., 2017*). It is possible that the included studies did not have enough time to develop muscle strength due to the shorter intervention periods. Age, gender, body mass index (BMI), playing level, injury history, data reporting, trainer blinding, control group exercises, screening tests, and player psychology might have impacted this systematic review. The average age of the participants ranged from thirteen to twenty-five, with the exception of one study (*Van Beijsterveldt et al., 2012*) that included older players and injuries are more common among older players. Because the likelihood of damage varied across sexes, gender played a role in the final results (*Doherty et al., 2014*). Risk of injury rises as body mass index (BMI) rises (*Alahmad et al., 2024*). The likelihood of injury is higher for recreational players compared to professionals (*Evans, 2003*; *Caine, Maffulli & Caine, 2008*). Recurrence is more likely in cases where injuries have occurred before (*Hale & Hertel, 2005*) and inaccuracies in reporting data might affect reliability (*Sadigursky et al., 2017*). Results were skewed since it was difficult to blind the trainers (*Willems et al., 2005*). There was a lack of direct comparison since the control group did not follow the same workout routine (*Owoeye et al., 2014*). Injury prevention and early detection might be possible using non-standard screening techniques (*Gabbe et al., 2004*). Lastly, the psychological state of players, though not addressed in the included studies, can significantly impact injury rates (*Ivarsson & Johnson, 2010*; *Ivarsson, Johnson & Podlog, 2013*).

The systematic review identified multiple clinical and contextual factors influencing injury outcomes. Age was a significant factor, as studies showed that older players are more prone to joint degeneration, impacting injury risk (*Soligard et al., 2008*). Specifically, *Van Beijsterveldt et al. (2012)* included players aged up to 40, which may have skewed results, as older players are more injury prone (*Van Beijsterveldt et al., 2012*). Gender also influenced findings, with studies indicating that male players face a higher general injury risk, while female players are more vulnerable to specific injuries like ankle sprains due to hormonal factors (*Alahmad et al., 2024*). Differences in BMI were noted, as higher BMI correlates with increased ankle injury risk due to joint wear (*Gribble et al., 2016*), though only two studies included BMI data (*Steffen et al., 2008*; *Tacconelli, 2010*). Playing level further impacted results, with amateur players, having lower skill levels, were more susceptible to injury than professionals (*Evans, 2003*; *Caine, Maffulli & Caine, 2008*).

A history of previous injuries was another risk factor, as players with prior injuries are at an 80% higher risk for recurrence (*Hale & Hertel, 2005*). Only one study excluded participants with recent injuries (*Owoeye et al., 2014*), potentially improving the reliability of its findings. Data reporting methods varied across studies, with inconsistencies in injury and exposure data collection leading to possible inaccuracies; for instance, *Soligard et al. (2008)* relied on volunteer reporting, which affected data accuracy. Trainer blinding was challenging, as the nature of exercise programs prevents placebo conditions (*Van Dyk, Behan & Whiteley, 2019*), and lack of blinding in all studies could introduce bias (*Willems et al., 2005*).

Control group exercise routines were often unspecified, with most control groups following "usual training" without details on components, making direct comparisons difficult (*Owoeye et al., 2014*). The inclusion of screening tests could further strengthen studies, as tests like the single-leg balance test can identify players at high risk of injury before the season starts (*Willems et al., 2005*). Lastly, the psychological state of players, such as anxiety or daily stress, is associated with injury risks in both male and female players (*Trojian & McKeag, 2006*). Monitoring mental health could improve injury prevention efforts.

The FIFA 11+ program demonstrated greater success in reducing ankle injury rates compared to its predecessor FIFA 11, likely due to its emphasis on neuromuscular control and progressive exercises. Variability in player attributes, such as age, gender, BMI, and playing level, alongside contextual factors like compliance rates, exposure hours, and intervention durations, significantly impacted outcomes. High compliance and direct supervision were associated with better results, though inconsistencies in control group routines and data reporting limited direct comparisons. Despite these challenges, the FIFA 11+ program showed promise in reducing ankle injuries.

## CONCLUSIONS

This systematic review provides evidence to support the effectiveness of the FIFA injury prevention programs, especially the FIFA 11+ program, which has already reduced ankle injuries in both male and female football players by up to 32%. Although there was no preventive effect on the incidence of ankle injuries when the data were pooled from the studies that used the FIFA 11 program (the old version) in the intervention group, the sport medicine organizations and football teams should nevertheless still be encouraged to implement the FIFA 11+ warm-up program as a safe and effective means to prevent football injuries in general and, in particular, to reduce the incidence of ankle injuries among football players.

### Limitations of the systematic review

This systematic review had several limitations that could have affected its overall findings. Firstly, it included only five studies, which is a small sample size potentially limited by the inclusion criteria. Secondly, the studies did not detail the traditional exercises used by the control groups, making it difficult to compare these with the FIFA injury prevention programs (FIFA 11 and FIFA 11+). Thirdly, only English-language studies were included, excluding potentially relevant studies in other languages. Fourthly, the review pooled data

from studies that focused on overall football injuries, with ankle injuries as a secondary outcome, lacking details on the types and severity of ankle injuries. Fifthly, a meta-analysis was not conducted due to participant and intervention heterogeneity. Finally, publication bias was not assessed because it is recommended only when ten or more studies are included (*Thorborg et al., 2017*).

## Future studies

To enable meaningful comparisons with the FIFA 11+ plan, future studies should define traditional training components in control groups. In order to compare FIFA 11+ to other injury prevention programs based on evidence, high-quality research is necessary. Additionally, it would be helpful for researchers to determine which activities help avoid football injuries and for how long, how hard, and how often. *Thorborg et al. (2017)* found that FIFA 11+ reduced injuries in amateur players, but it has to be studied more in professional players to determine its usefulness.

## Implications for clinical practice

Although playing football regularly has many positive effects on health, it also has a high risk of injury. Injuries are more common, particularly among young people, when there are issues like unsuitable playing grounds, insufficient exercise facilities, and a lack of education on how to prevent injuries (*Owoeye et al., 2014*). In settings with limited resources, it is especially crucial to implement programs like FIFA 11+ that are backed by evidence in order to decrease injuries and the corresponding medical expenses (*Al Attar et al., 2016*; *Tacconelli, 2010*).

The results of this research illustrate the practical advantages of implementing the FIFA 11+ program into regular football training to decrease ankle injuries. Coaches may utilize its planned warm-up activities to help players improve their neuromuscular control, stability, and proprioception, all of which are important variables in avoiding ankle sprains during high-impact actions. For best results, adhere to the regimen at least twice a week. FIFA 11+'s modest equipment needs make it a viable injury prevention method for resource-limited and amateur contexts, promoting safer and more sustainable football participation at all levels.

### Funding
This research was funded by AlMaarefa University, Riyadh, Saudi Arabia. The funders had no role in study design, data collection and analysis, decision to publish, or preparation of the manuscript.

### Grant Disclosures
The following grant information was disclosed by the authors:
AlMaarefa University, Riyadh, Saudi Arabia.

### Competing Interests
The authors declare there are no competing interests.

## Author Contributions

- Mohammad Alhazmi conceived and designed the experiments, performed the experiments, prepared figures and/or tables, authored or reviewed drafts of the article, and approved the final draft.
- Emad Alhazmi performed the experiments, prepared figures and/or tables, authored or reviewed drafts of the article, and approved the final draft.
- Wael Abdulghani Alghamdi performed the experiments, prepared figures and/or tables, authored or reviewed drafts of the article, and approved the final draft.
- Mohammed Zalah conceived and designed the experiments, performed the experiments, prepared figures and/or tables, authored or reviewed drafts of the article, and approved the final draft.
- Shadab Uddin performed the experiments, analyzed the data, authored or reviewed drafts of the article, and approved the final draft.
- Moattar Raza Rizvi conceived and designed the experiments, analyzed the data, authored or reviewed drafts of the article, and approved the final draft.
- Fuzail Ahmad conceived and designed the experiments, analyzed the data, authored or reviewed drafts of the article, and approved the final draft.

## Data Availability

The systematic review synthesizes data from multiple studies that report aggregate outcomes rather than individual-level data. As a result, the raw data is not directly available in a usable form for sharing.

## Supplemental Information

Supplemental information for this article can be found online at http://dx.doi.org/10.7717/peerj.18910#supplemental-information.

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
