# Peer review of "Effectiveness of FIFA injury prevention programs in reducing ankle injuries among football players: a systematic review"

_PeerJ, doi:10.7717/peerj.18910_

## Round 0.1 · original submission · Major Revisions

Dear Authors:

Thank you for submitting your manuscript titled "Effectiveness of FIFA injury prevention programs in reducing ankle injuries among football players: a systematic review". After evaluating the reviews, we will consider major reviews.

Please attend to the reviewer´s comments.

Regards

Dr. Manuel Jiménez

Reviewer 1 ·

Basic reporting

Dear Authors,
thank you for your interesting work. Although I appreciated it, I would like to read a meta-analysis to receive stronger and more appropriate results. However, the English is clear, unambiguous and correct. A sufficient background has been stated. The structure is professional according to scientific review.

Experimental design

The research question is clear and the useful applications have been well commented. A rigorous investigation has been conducted, but I believe the rationale for preferring a systematic review rather than a meta-analysis should be better justified. Can you estimate a heterogeneous index that led you to exclude that experimental design?
Also, why did you not consider the grey literature too?
However, the other methods have been clearly described.

Validity of the findings

Although the findings' validity and discussion debated all the results, many statements need a meta-analysis and the ES estimation for their justification. It is not possible to conclude all the statements you reported. In addition, I believe that the discussion section may be improved including your interpretation and rationale that follows the results obtained. It appears as a list of previous research without actual investigators' contributions.

Reviewer 2 ·

Basic reporting

See below

Experimental design

See below

Validity of the findings

See below

Additional comments

Many thanks to the authors for presenting such an interesting systematic review about “Effectiveness of FIFA injury prevention programs in reducing ankle injuries among football players”.

Before resubmitting the revision version of the article, please read the editorial rules carefully, and check other editorial aspects.
The article is generally well-written and uses adequate professional language. However, there are some minor issues with sentence structure and clarity that should be addressed to enhance readability, especially in the "Introduction" section. For example, sentences such as, “The ankle joint is the most susceptible area for football injuries,” could be more precise by specifying the most common injury mechanism in football for the ankle.

Title and Abstract
The title is appropriate, concise and captivating.
The abstract is well structured, it contains the main information of the study and presents it in an adequately brief way.

Key words
Please list them in alphabetical order.

Background
The introduction effectively contextualizes the significance of football injuries and expecially ankle injuries, but it could benefit from a brief comparison to non-soccer sports where such lesions are also prevalent. The authors highlight several key statistics (such as the costs of treatment in different countries) which are useful, but they do not sufficiently frame the scientific rationale for specifically focusing on the effectiveness of FIFA 11+ and FIFA 11. More detail could be provided on the unique aspects of these programs compared to other injury prevention strategies (plyometric training, proprioceptive exercises).
Lines 37-47 please add the role of Kinesio Taping in ankle joint performance, quoting:
https://pubmed.ncbi.nlm.nih.gov/35630037/

Methods
The authors followed standard guidelines for systematic reviews, but the methodology could be more rigorous in several areas: Although the authors searched five databases, the time period (2008–2024) and manual screening methods may have introduced bias. The inclusion of only randomized controlled trials (RCTs) is sound, but the exclusion of cohort studies (often used in sports injury research) limits the scope of the review. The use of the PEDro scale to assess methodological quality is appropriate, but the quality scores (ranging from 5–7) indicate that some studies might not meet the highest standards of rigor. It would be helpful to explain which specific methodological weaknesses (e.g., lack of blinding or inadequate follow-up) affected these scores.

Results
The results section provides useful data on the effectiveness of the FIFA 11+ and FIFA 11 programs in reducing ankle injuries but has several key limitations. There is insufficient analysis of why FIFA 11 is less effective than FIFA 11+, despite a higher injury rate with the older program. Additionally, the differences between participant demographics (age, gender, skill levels) across studies are mentioned but not fully explored. The section also lacks details on the control groups' usual training programs, making comparisons difficult

Discussion
The discussion appropriately focuses on the success of FIFA 11+ but doesn't address why this program is more effective than FIFA 11 in detail. The conclusion that FIFA 11+ is particularly effective should also discuss whether these findings can be generalized to all populations, especially given the lack of professional players in the included studies.
There is, however, a critical lack of analysis regarding the compliance issue. For example, compliance ranged from 46% to 77% in different studies, and it is unclear whether lower compliance explains the discrepancies in results. Some reasoning on this point could increase the quality of the discussion.
Discuss your data with those presented in literature regarding different anatomical district

Conclusions
The conclusion section is adherent to the results presented.

References
The references listed appear up to date and adequately selected, but they should be integrated as suggested previously.

Tables and Figures
The number and quality of tables and figures are appropriate to transmit the main information of the paper.

·

Basic reporting

1. Clear and unambiguous, professional English used throughout.
The article is generally written in clear and professional English, but there are a few areas where clarity could be improved. Some sentences are overly complex or redundant, which may hinder readability for an international audience. For example, in the abstract, the sentence “can lead to preventing players from participating in sport” could be simplified to “can prevent players from participating in sport.”

Suggested Improvement: Consider a professional English editing service or a colleague proficient in English to refine the language for clarity and fluidity, especially in complex sentences.

2. Literature references, sufficient field background/context provided.
The article provides sufficient background and a solid review of relevant literature. It covers key studies on FIFA 11 and FIFA 11+ injury prevention programs and discusses their relevance to ankle injuries in football. However, a broader discussion of injury prevention in sports, beyond FIFA programs, could enhance the context and provide a more comprehensive background for readers unfamiliar with these programs.

Suggested Improvement: Incorporate more references to alternative injury prevention strategies in sports to provide a broader context for the study.

3. Professional article structure, figures, tables. Raw data shared.
The structure of the article conforms to academic standards, with appropriate use of headings, figures, and tables. The tables effectively summarize data and are easy to interpret. However, the description of the PRISMA flow diagram could be expanded to provide more detail about the selection process for the studies included in the review.

Suggested Improvement: Add more detail to the description of Figure 1 (PRISMA flow diagram) to explain the study selection and exclusion process in greater depth.

4. Self-contained with relevant results to hypotheses.
The article is self-contained and presents results that are directly relevant to the research hypotheses. It clearly links the effectiveness of the FIFA 11+ program to the reduction of ankle injuries, while also addressing the limitations of the older FIFA 11 program. However, the connection between the findings and the broader implications for injury prevention in football could be more thoroughly explored.

Suggested Improvement: Expand the discussion on the practical implications of the findings, especially for coaches and sports practitioners, and how they can apply the results to reduce ankle injuries in football.

5. The introduction part is written in great detail, but ankle injuries and ankle kinesiology are not mentioned much. A reference to ankle kinesiology would be appropriate.

Experimental design

1. Original primary research within Aims and Scope of the journal.
The article adheres to the aims and scope of the journal, as it focuses on injury prevention in football, a key area of sports medicine and rehabilitation. The systematic review of FIFA injury prevention programs (FIFA 11 and FIFA 11+) in reducing ankle injuries aligns well with the journal’s goal of publishing research that contributes to understanding injury prevention, treatment, and rehabilitation strategies in sports. By focusing on ankle injuries—a common and serious issue in football—the article addresses a timely and relevant problem that fits the journal's focus on injury prevention.
2. Research question well defined, relevant & meaningful.
The research question—whether FIFA injury prevention programs are effective in reducing ankle injuries is clearly defined, relevant, and significant within the field. Ankle injuries are prevalent in football, and understanding how specific injury prevention programs can reduce their occurrence is of great interest to sports professionals and researchers. The study effectively addresses a gap in the literature, as previous research has focused on overall injury prevention but has not thoroughly examined the specific impact on ankle injuries.
3. Rigorous investigation performed to a high technical & ethical standard. The systematic review follows a rigorous methodology, including the registration of the review protocol in PROSPERO and the use of the PEDro scale to assess the quality of included studies. The inclusion of RCTs strengthens the scientific rigor of the review, ensuring that the findings are based on high-quality evidence. However, the study acknowledges the heterogeneity in participant characteristics and interventions across studies, which could introduce bias.
Suggested Improvement: While the study appropriately notes the limitations due to heterogeneity, it would benefit from a more detailed discussion of how this heterogeneity was managed in the analysis, such as considering subgroup analyses based on participant age, gender, or playing level.
4. Methods described with sufficient detail & information to replicate. The methodology is described with sufficient detail, including the search strategy, databases used, inclusion criteria, and the process of study selection. However, the description of the control groups’ training regimens in the included studies is insufficient, which makes it difficult to understand the full context of the intervention comparisons.
Suggested Improvement: The manuscript should provide more information about the control groups' training programs, as this would enable a clearer comparison between the intervention and control conditions. If such data are unavailable, it should be more explicitly addressed as a limitation.

Validity of the findings

1. Impact and novelty not assessed. Meaningful replication encouraged where rationale & benefit to literature is clearly stated.
The article does not explicitly discuss the impact or novelty of its findings within the broader field of sports injury prevention. While the study fills a gap in the literature by focusing on ankle injuries and the FIFA 11+ program, it could further emphasize the novelty of this focus and how it advances current knowledge.

Suggested Improvement: Highlight the novelty of the research more explicitly in the discussion, especially in relation to previous systematic reviews and meta-analyses that primarily focused on general injury prevention. Clarify how the findings contribute new insights to the literature and encourage future research to replicate the findings in different populations or settings.

2. All underlying data have been provided; they are robust, statistically sound, & controlled.
The article provides a thorough review of the underlying data, particularly through the use of the Physiotherapy Evidence Database (PEDro) scale to assess the quality of included studies. The data from the RCTs are robust and statistically sound. However, there is variability (heterogeneity) in the included studies, such as differences in participants' age, gender, and playing levels, which may influence the outcomes.

Suggested Improvement: While the article acknowledges heterogeneity, it would benefit from discussing it in more detail. Consider providing sensitivity analyses or subgroup analyses to explore how this variability might affect the robustness of the findings. Addressing how this heterogeneity was controlled in the analysis would strengthen the study’s validity.

3. Conclusions are well stated, linked to original research question & limited to supporting results.
The conclusions are well articulated and clearly linked to the original research question, especially in terms of the effectiveness of the FIFA 11+ program in reducing ankle injuries. The authors appropriately limit their conclusions to the results presented in the included studies. However, the discussion around the older FIFA 11 program could be expanded to offer more insights into why it failed to reduce ankle injuries compared to the FIFA 11+ program.

Suggested Improvement: Expand the conclusions to provide a more in-depth discussion of the potential reasons behind the differing effectiveness of FIFA 11 and FIFA 11+. This could help practitioners understand how and why the FIFA 11+ program works better, leading to more informed decisions in injury prevention strategies.

In the study, as well as in the other included studies, the method used to determine ankle sprain injuries has not been clearly explained, nor have the injuries been classified according to severity. This is a significant limitation as the lack of a standardized injury definition and grading system can affect the accuracy and consistency of the findings. Future studies should ensure that clear criteria are used to define injuries and that the severity of each injury is properly categorized to improve the reliability of the results.

Additional comments

Areas for Improvement:

Lack of Injury Definition and Classification: One notable limitation is the lack of clarity on how ankle sprains were defined across studies and the absence of injury severity grading. This omission can lead to inconsistencies in the interpretation of results, as different studies may have used varying definitions for what constitutes an ankle injury. Providing a standardized definition and classification system for injuries would enhance the study’s reliability.
Heterogeneity Between Studies: There is substantial heterogeneity in the included studies, especially regarding participants' age, gender, and playing levels, as well as differences in the interventions (FIFA 11 vs. FIFA 11+). While this heterogeneity is acknowledged, a more thorough exploration of its potential impact on the results would strengthen the paper.
Limited Discussion on FIFA 11 Program: While the article clearly demonstrates the effectiveness of FIFA 11+ in reducing ankle injuries, the failure of FIFA 11 to produce similar results deserves more detailed analysis. Understanding why FIFA 11 was less effective could offer important lessons for improving future injury prevention programs.
Language Clarity: Though the article is well-written, there are areas where the language could be further refined for clarity. Some sentences are unnecessarily complex, and simplifying these would improve readability.

Search Strategy: The search strategy includes five major databases, which is sufficient for comprehensiveness. The search terms and strategy are clearly defined. However, the exclusion of non-English studies could introduce language bias, which should be acknowledged.

---

## Round 0.2 · Minor Revisions

Dear Authors:

Reviewer #2 asks for a new comparative and a new citation to your manuscript. Don´t forget that you are not obligated to introduce a citation if you do not agree with the recommendation. In any case, I agree to accept your paper after you consider the reviewer's comments or explain why you do not agree with them.

Thank you,

Dr. Manuel Jiménez

Reviewer 1 ·

Basic reporting

The authors improved the manuscript following the reviewer's suggestions. In my opinion, no more effort is needed.

Experimental design

The authors improved the manuscript following the reviewer's suggestions. In my opinion, no more effort is needed.

Validity of the findings

The authors improved the manuscript following the reviewer's suggestions. In my opinion, no more effort is needed.

Reviewer 2 ·

Basic reporting

OK

Experimental design

OK

Validity of the findings

OK

Additional comments

Please, in the discussion section, as the authors intend to emphasize the FIFA program, I invite them to make a comparison with other programs in use for injury prevention, quoting:
- https://pubmed.ncbi.nlm.nih.gov/34444026/

---

## Round 0.3 · accepted · Accept

Dear Coauthors:

Your manuscript, “Effectiveness of FIFA Injury Prevention Programs in Reducing Ankle Injuries Among Football Players: A Systematic Review,” showed relevance, was well-presented, and provided valuable insights into the effectiveness of FIFA injury prevention programs. While the manuscript could benefit from including more recent studies (post-2020) and providing clearer descriptions of control group interventions and study heterogeneity, these limitations do not detract from its overall quality and impact.

Congratulations,

Dr. Manuel Jiménez

Reviewer 2 ·

Basic reporting

OK

Experimental design

OK

Validity of the findings

OK

Additional comments

I appreciate the authors' efforts to address my concerns and update their manuscript. Although my opinion about the depth of the study remains the same, I am now satisfied with the changes made.

·

Basic reporting

Self-contained with relevant results to hypotheses.

Experimental design

Methods described with sufficient detail & information to replicate.

Validity of the findings

Conclusions are well stated, linked to original research question & limited to supporting results.

Additional comments

Dear Editor,

The revised manuscript titled "Effectiveness of FIFA Injury Prevention Programs in Reducing Ankle Injuries Among Football Players: A Systematic Review" indicates that the authors have addressed the majority of the requested revisions. While the manuscript is in a publishable state, some shortcomings remain.

Revision Status:
The authors have adequately revised the manuscript, and the core objectives of the study have been clearly articulated and addressed. The findings are well-presented and offer valuable insights into the effectiveness of FIFA injury prevention programs in reducing ankle injuries.

Literature Update:
A notable limitation is the lack of integration of more recent literature. Studies published after 2020 are underrepresented, which may affect the comprehensiveness of the review. Given the specificity of the topic, this could be partly understandable; however, including more recent studies would strengthen the manuscript and enhance its relevance to current practices.

Remaining Issues:

The control group interventions remain inadequately described, making it difficult to compare them effectively with the FIFA injury prevention programs.
While heterogeneity among the included studies has been acknowledged, a deeper analysis of these variations (e.g., participant demographics, intervention protocols, and compliance levels) would provide further clarity and context for the findings.
Recommendation:
Despite these limitations, the manuscript provides significant value to the field of sports injury prevention. The authors have made meaningful revisions, and the study’s focus on a specific type of injury prevention is both relevant and impactful.

In conclusion, I recommend the manuscript for publication, with the suggestion that the authors explicitly acknowledge the limited representation of post-2020 literature as a limitation in the final version.

Sincerely.